# Entrepreneurship Research in Times of COVID-19: Experiences from South America

**Víctor Hugo Fernández-Bedoya \*, Monica Elisa Meneses-La-Riva** **, Josefina Amanda Suyo-Vega and Johanna de Jesús Stephanie Gago-Chávez**

Grupo de Investigación Sostenibilidad, Universidad César Vallejo, Av. Alfredo Mendiola, Trujillo 13600, Peru
\* Correspondence: vfernandezb@ucv.edu.pe

**Abstract:** The pandemic caused by COVID-19 has had diverse effects on the population and businesses. Due to their low visibility, there is a significant knowledge gap for the scientific community regarding the situation of South American entrepreneurship during the COVID-19 pandemic. Therefore, this systematic review aims to answer the following research questions: What scientific evidence is available on entrepreneurship in South America during the COVID-19 pandemic? From which countries do the studies originate? What conclusions do they present, and what lessons can be learned from them? The Scopus and Scielo databases were explored in this systematic review. Due to the diversity of languages in which articles from this region are published, searches were conducted in English, Spanish, and Portuguese. In each case, we searched for results containing the words COVID-19 and entrepreneurship (*emprendimiento* in Spanish, *empreendedorismo* in Portuguese) in the title, abstract, or keywords. The initial search produced 783 records, which were filtered based on seven criteria, resulting in 15 articles. The final articles identified had corresponding authors affiliated with South American institutions. We conclude that, in South America, some entrepreneurs recognize the importance of maintaining the economic stability of their employees and communities in addition to sustaining their businesses. Their actions serve as case studies of resilience and perseverance in adverse circumstances.

**Keywords:** entrepreneurship; COVID-19; business ventures; employment; adverse scenario; South America; systematic review

## 1. Introduction

One tool for assessing a country's economic growth is by analyzing its GDP (gross domestic product) per capita growth, which is the currency value of the total goods and services produced in a country during a given period, divided by the number of inhabitants [1–3]. This measure provides a balanced scorecard for evaluating a country's economic health [4–6]. To calculate a country's GDP per capita, one must consider private and public consumption, government outlays, investments, additions to private inventories, construction costs paid, and foreign trade balance, all divided by the number of inhabitants [7,8].

Previous research has shown that regions with high levels of entrepreneurship and business innovation tend to experience higher levels of economic growth and development [9,10]. This reaffirms Joseph Schumpeter's assertion more than a hundred years ago that "the company and the entrepreneur are the engine of economic development" [11]. It is important to note that innovation is a necessary ingredient for breaking static economies and establishing business development led by entrepreneurship as the basis for economic growth [9].

Perhaps the most widely accepted definition of entrepreneurship is the one put forth by [12], which describes it as the capacity and willingness of individuals to perceive and create new economic opportunities, such as new goods or services, methods of production, organizational schemes, and other valuable intellectual creations. This often involves

introducing their ideas to a market full of obstacles and uncertainty, making decisions about the form, place, and use of their creation, and generating demand from market players [12,13].

Entrepreneurship emerged during the interwar period of the 20th century as a strategy developed by the North American and European governments to mitigate the unemployment that prevailed at the time [14]. Today, governments around the world seek to promote a culture of entrepreneurship as a means of job creation and competitiveness, contributing to social welfare and governance by increasing citizens' income and resulting in positive variations in GDP per capita [15–18]. However, for a venture to be sustained over time, formalized, and generate income for the state through tax payments, it requires planning skills and resilience [19,20]. Although ventures usually start on a small scale, they can become competitive international companies by developing an international entrepreneurial orientation in their leaders [21,22].

Entrepreneurship is critical to economic development and progress [23]. Entrepreneurs generate innovation, job creation, and economic progress [23,24]. They uncover new market possibilities, produce new goods and services, and generate new employment, all of which contribute to higher productivity and economic growth [25]. Entrepreneurship also encourages competition, which helps to lower prices and increase quality.

The institutional environment, such as government policies, laws, regulations, and social norms, can significantly influence entrepreneurship and economic development [26,27]. For example, a supportive institutional environment can encourage entrepreneurs to start and grow businesses by providing access to finance, protecting property rights, and creating a favorable business environment [28]. In contrast, an unfavorable institutional environment can discourage entrepreneurship by making it difficult to start and operate a business, limiting access to finance, and creating barriers to entry [26,29].

Furthermore, the institutional environment might have an impact on the sort of entrepreneurship that arises in a certain nation or area [30–32]. Certain institutional contexts, for example, may be more suited to high-growth entrepreneurship, whereas others may be better suited to informal entrepreneurship [33–38]. Rapid development, innovation, and job creation describe high-growth entrepreneurship, whereas small-scale, low-growth operations, frequently in the informal sector, characterize informal entrepreneurship [39–41].

Ultimately, the institutional framework is crucial in encouraging entrepreneurship and economic growth. An institutional environment that promotes entrepreneurship may lead to greater innovation, job creation, and economic growth, whereas an unfavorable institutional environment can stifle entrepreneurship and impede economic progress [42,43].

The scientific literature from South America has made major contributions to the research on COVID-19 and its influence on the area [44–47]. Researchers have investigated how Latin American nations might adapt to the epidemic and advance toward economic success by focusing on entrepreneurship and innovation [48].

One critical subject addressed in South American scientific literature is the role of entrepreneurship in boosting economic growth during the epidemic [49]. Many studies have shown the role of SMEs in stimulating economic activity, creating jobs, and promoting innovation [50,51]. According to studies, firms with a high level of entrepreneurship are more robust and flexible during crises such as the COVID-19 epidemic [49,52,53].

South American scientific literature has investigated the influence of government policies and programs in supporting SMEs and encouraging entrepreneurship [54]. Researchers have investigated the efficacy of numerous efforts, such as financial aid programs, tax breaks, and regulatory changes [55,56]. They have also considered the significance of establishing networks and collaborations among SMEs, universities, and other stakeholders in order to foster innovation and knowledge-sharing [57–61].

Another major topic addressed in South American scientific literature is the influence of COVID-19 on vulnerable populations, such as the poor [62–66]. According to studies, tailored policies and actions are needed to help these groups and alleviate the economic and social repercussions of the epidemic. Researchers have investigated the efficacy of different

initiatives, such as cash transfers and food aid, in decreasing poverty and promoting economic resilience [67–71].

Finally, South American scientific literature has emphasized the importance of collaboration and knowledge-sharing in addressing the challenges of COVID-19 [72]. Researchers have highlighted the need for international cooperation and the sharing of data, expertise, and resources to develop effective strategies for responding to the pandemic. They have also emphasized the importance of interdisciplinary research and collaboration between scientists, policymakers, and entrepreneurs to address the complex economic and social issues raised by the pandemic.

Research related to identifying the factors that motivate or reinforce the interest in entrepreneurship has been conducted due to the demand for knowledge related to the topic by the state and companies [73–82].

The COVID-19 pandemic has provided significant lessons for the global business community [83–87]. It has resulted in the bankruptcy of small businesses, leading to job losses or reduced working hours [88–91], and shorter working hours have led to lower wages [92,93] due to the confinement measures implemented by governments [94–98]. Only robust companies with good strategic planning have been able to continue hiring staff during this adverse scenario [99–104].

The enterprises that remained in operation not only contributed to the continuity of local development but also provided essential goods and services to the population in need. Therefore, some authors consider entrepreneurship as the foundation for the growth and development of society [105,106].

There are gaps in the literature concerning entrepreneurship in South America. In particular, it is necessary to identify the motivations and experiences of the entrepreneurs who have survived this challenging phase, such as the COVID-19 pandemic, with a focus on South America.

Therefore, this research had as research questions: What scientific evidence is available on entrepreneurship in South America during the COVID-19 pandemic? From which countries do the studies originate? What conclusions do they present, and what lessons can be learned from them?

In order to answer these research questions, a systematic review of the specialized literature in the Scopus database was proposed. Due to the diversity of authors in this geographical area, scientific evidence published in English, Spanish, and Portuguese was included.

## 2. Materials and Methods

This study is a systematic review. It is characterized by an orderly and explicit evaluation of the literature based on a clear research question, together with a critical analysis of the evidence found [107].

The searches were first conducted in Scopus, which is considered by the academic community as a high-quality database, in addition to its broad coverage of publications in the social sciences and humanities [108]. Due to the small number of results identified, it was also decided to use the Scielo database. Scielo is the highest quality regional database in Ibero-America, which is constantly nourished by contributions in Spanish and Portuguese [109].

Since the predominant languages in the scientific literature from South America are Spanish, Portuguese, and English, it was necessary to carry out searches in these languages. In each case, we searched for results containing the words COVID-19 and entrepreneurship (*emprendimiento* in Spanish, *empreendedorismo* in Portuguese) in the title, abstract, or keywords, as detailed in Table 1. It is important to emphasize that the searches were conducted on 1 July 2022 (in Scopus) and 15 November 2022 (in Scielo). This was because it was originally planned to search only in Scopus, but due to the small number of records identified, it was later decided to use the Scielo database as well.

**Table 1.** Searches.

| Code | Search | Initial Results |
|:---:|:---:|:---:|
| A1 | TITLE-ABS-KEY (entrepreneurship AND COVID-19) | 746 |
| A2 | TITLE-ABS-KEY (*emprendimiento* AND COVID-19) | 25 |
| A3 | TITLE-ABS-KEY (*empreendedorismo* AND COVID-19) | 12 |

Then, we applied filters to refine the search results according to seven criteria:

- Criteria 1: Temporality. Results were sought in the context originated by COVID-19. We sought to exclude research results published before 2019 that might have infiltrated the initial results. This process did not remove any articles, so the partial results remained at 783 records.
- Criteria 2: Original article. We sought to identify evidence of entrepreneurship that is disclosed through original articles. We excluded other types of articles, such as literature reviews, letters to the editor, etc. In this process, 208 articles were excluded, resulting in 575 records.
- Criteria 3: Country of origin. We eliminated all articles that were registered as the geographical context in countries other than South America. This process eliminated a large number of records (481), reducing the partial result to 94.
- Criteria 4: Language. Due to the diversity of languages spoken in South America, English, Spanish, and Portuguese were identified as the most commonly used languages for the communication of scientific results. Therefore, we sought to remove any scientific article that was not written in these languages. No articles with these characteristics were identified, so the partial result remained at 94 records.
- Criteria 5: Full access. To examine the pre-selected articles, the feasibility of their free download from Scopus was verified. The Universidad César Vallejo provides us with institutional access to this database; however, due to editorial policies, some of the articles identified could not be freely accessed. In this process, 31 records were discarded, reducing the partial results to 63.
- Criteria 6: Duplicity. Since we performed 3 different searches (coded as A1, A2, and A3), we feared that there might be duplicate records. This did happen. We were surprised to find that all of the records found in searches A2 and A3 (in Spanish and Portuguese, respectively) were identified in search A1 (in English). This is due to the current trend of South American journals presenting both abstract and *resumen* (abstract in Spanish) or abstract and *resumo* (abstract in Portuguese). The metadata captures information in both languages for greater visibility. As a result of this process, 19 articles were withdrawn, and 44 partial results were counted.
- Criteria 7: Relevance (full-text review). The individual reading of each scientific article made it possible to define whether or not they answered the research question. We found some articles that were systematic reviews and opinion articles cataloged as original articles, which is an error in the metadata. We also identified some articles on global analysis of economies with little or no emphasis on South America, which was also removed from the analysis. This thorough analysis resulted in the removal of 29 articles, resulting in the identification of 15 records that meet the research objectives.

The complete process can be reviewed in Table 2. In addition, Figure 1 presents the Preferred Reporting Items for Systematic Review and Meta-Analyses (PRISMA) flow chart [110].

**Table 2.** Criteria followed in the process of obtaining final results.

| Criteria Followed | A1 | A2 | A3 | Total |
|---|---|---|---|---|
| Initial results | 746 | 25 | 12 | 783 |
| Criteria 1: temporality | 0 | 0 | 0 | 0 |
| Partial results | 746 | 25 | 12 | 783 |
| Criteria 2: original article | 207 | 1 | 0 | 208 |
| Partial results | 539 | 24 | 12 | 575 |
| Criteria 3: country of origin | 479 | 2 | 0 | 481 |
| Partial results | 60 | 22 | 12 | 94 |
| Criteria 4: language | 0 | 0 | 0 | 0 |
| Partial results | 60 | 22 | 12 | 94 |
| Criteria 5: Full access | 22 | 7 | 2 | 31 |
| Partial results | 38 | 15 | 10 | 63 |
| Criteria 6: Duplicity | 10 | 7 | 2 | 19 |
| Partial results | 28 | 8 | 8 | 44 |
| Criteria 7: Relevance (full text review) | 22 | 4 | 3 | 29 |
| Final results | 6 | 4 | 5 | 15 |

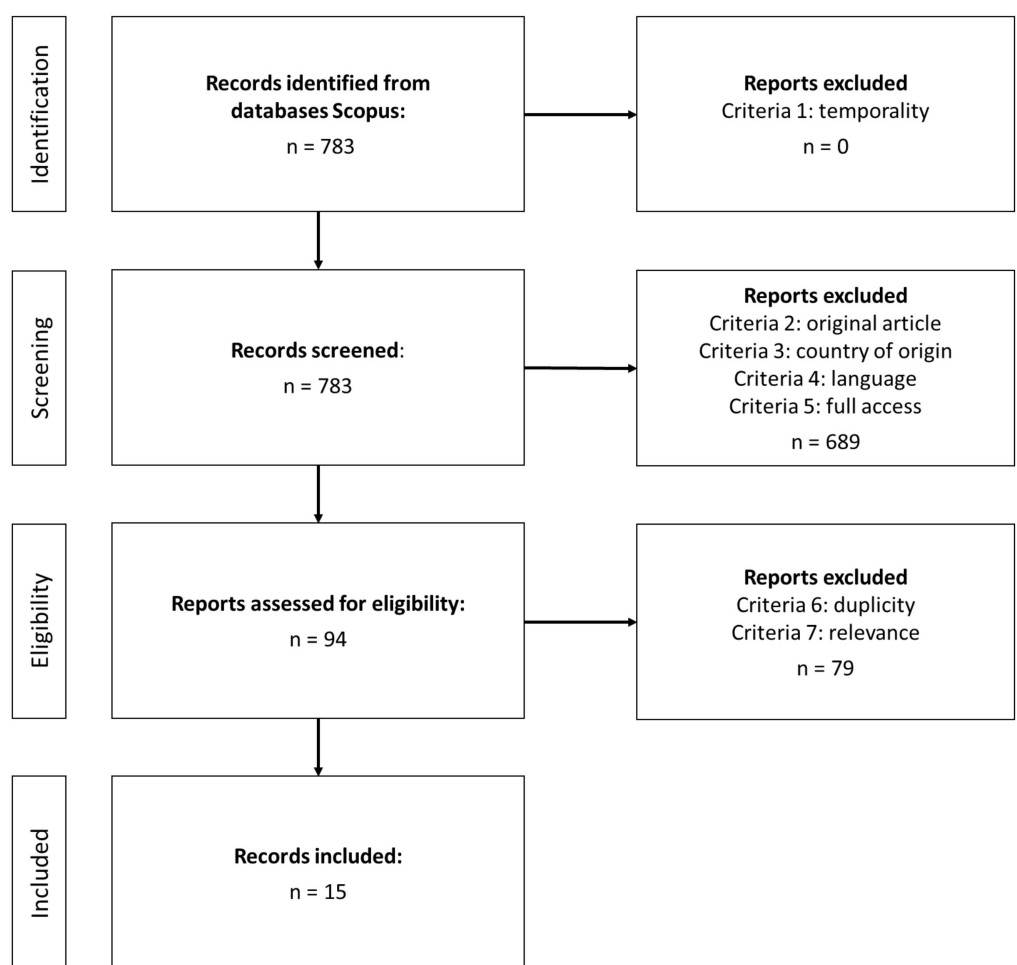

**Figure 1.** Prisma flow chart.

## 3. Results

The six records identified are presented in Table 3. It shows the internal code assigned to each of them, the citation, the original title, and the scientific journal where it was published.

**Table 3.** Articles identified.

| Code | Citation | Title | Journal |
|---|---|---|---|
| A1-1 | [111] | Ideas de negocio para reactivar el turismo en pandemia en "La Encantada", Chulucanas, Perú [Business ideas to reactivate tourism in the pandemic in "La Encantada", Chulucanas, Peru] | Revista Venezolana de Gerencia |
| A1-2 | [112] | Entrepreneurial Nursing interventions for the social emancipation of women in recycling | Revista da Escola de Enfermagem |
| A1-3 | [113] | Job autonomy, unscripted agility and ambidextrous innovation: analysis of Brazilian startups in times of the COVID-19 pandemic | Revista de Gestão |
| A1-4 | [114] | As configurações do trabalho musical e a pandemia da COVID-19: precarização, luto, resiliência e redes de cooperação [The configurations of musical work and the COVID-19 pandemic: precarization, mourning, resilience and networks] | Opus |
| A1-5 | [115] | Relacional: Easing the Crisis Effects in the Education Sector | Revista de Administração Contemporânea |
| A1-6 | [116] | Workplace Situation and Well-Being of Ecuadorian Self-Employed | Sustainability |
| A2-1 | [117] | Importancia de las TIC en circuitos cortos de comercialización de alimentos [Importance of ICTs in short food marketing circuits] | RIVAR |
| A2-2 | [118] | Peruana del bicentenario: promotora del emprendimiento en tiempos de crisis [Peruana del bicentenario: Promoting entrepreneurship in times of crisis] | Comunicción: Revista de Investigación en Comunicación y Desarrollo |
| A2-3 | [119] | Emprendimiento y resiliencia: caso de las bodegas de barrio en el Perú durante la pandemia de COVID-19 [Entrepreneurship and resilience: the case of neighborhood bodegas in Peru during the COVID-19 pandemic] | Desde el Sur |
| A2-4 | [120] | Emprendimiento en tiempo de crisis: una evaluación al impacto del COVID en las PYMES de la Provincia de El Oro, Ecuador [Entrepreneurship in times of crisis: an evaluation of the impact of COVID on SMEs in the Province of El Oro, Ecuador] | Dilemas contemporáneos: Educación, Política y Valores |
| A3-1 | [121] | Digital transformation in university technology expo | Revista de Administração Mackenzie |
| A3-2 | [122] | Impactos da pandemia de COVID-19 sobre o empreendedorismo digital nas instituições bancárias brasileiras: uma análise à luz das forças isomórficas [Impacts of the COVID-19 pandemic on digital entrepreneurship in Brazilian banking institutions: an analysis in the light of isomorphic forces] | Estudios Gerenciales |
| A3-3 | [123] | Empreendedorismo e coronavírus: impactos, estratégias e oportunidades frente à crise global [Entrepreneurship and coronavirus: impacts, strategies and opportunities facing the global crisis] | Estudios Gerenciales |
| A3-4 | [124] | Stay at Home, Casa Porto Delivers: Humanized Entrepreneurship during Pandemic | Revista de Administração Contemporânea |
| A3-5 | [125] | Wanderlust without Wandering: Managing a Travel Blog during the COVID-19 | Revista de Administração Contemporânea |

Due to the wide diversity existing in South America, it was necessary to specify the original language of each scientific article, as well as the geographic context from which the evidence of business ventures was extracted. In some cases, the authors focused on a specific city (A1-1, A1-4, A1-6, A2-1, A2-2, A2-4, A3-1, A3-3, A3-4), in one case, a region (A1-2), and in the remaining cases the entire country (A1-3, A1-5, A2-3, A3-2, A3-5). This is detailed in Table 4.

**Table 4.** Language and geographical context for each article identified.

| Code | Language | Geographical Context |
|------|----------|---------------------|
| A1-1 | Spanish | Piura, Peru |
| A1-2 | English | Southern Brazil |
| A1-3 | English | All of Brazil |
| A1-4 | Portuguese | Curitiba, Brazil |
| A1-5 | English | All of Brazil |
| A1-6 | English | Manabí, Ecuador |
| A2-1 | Spanish | Mar del Plata, Argentina |
| A2-2 | Spanish | Tacna, Peru |
| A2-3 | Spanish | All of Peru |
| A2-4 | Spanish | El Oro, Ecuador |
| A3-1 | Portuguese (English version available) | Blumenau, Brazil |
| A3-2 | Portuguese | All of Brazil |
| A3-3 | Portuguese | Paraná, Santa Catarina and Rio Grande do Sul, Brazil |
| A3-4 | Portuguese (English version available) | Rio de Janeiro, Brazil |
| A3-5 | Portuguese (English version available) | All of Brazil |

We considered that it was necessary to present a figure that presents the South American countries to the international public. Figure 2 presents such a graphic, in addition to the identification of the origin of the six records identified.

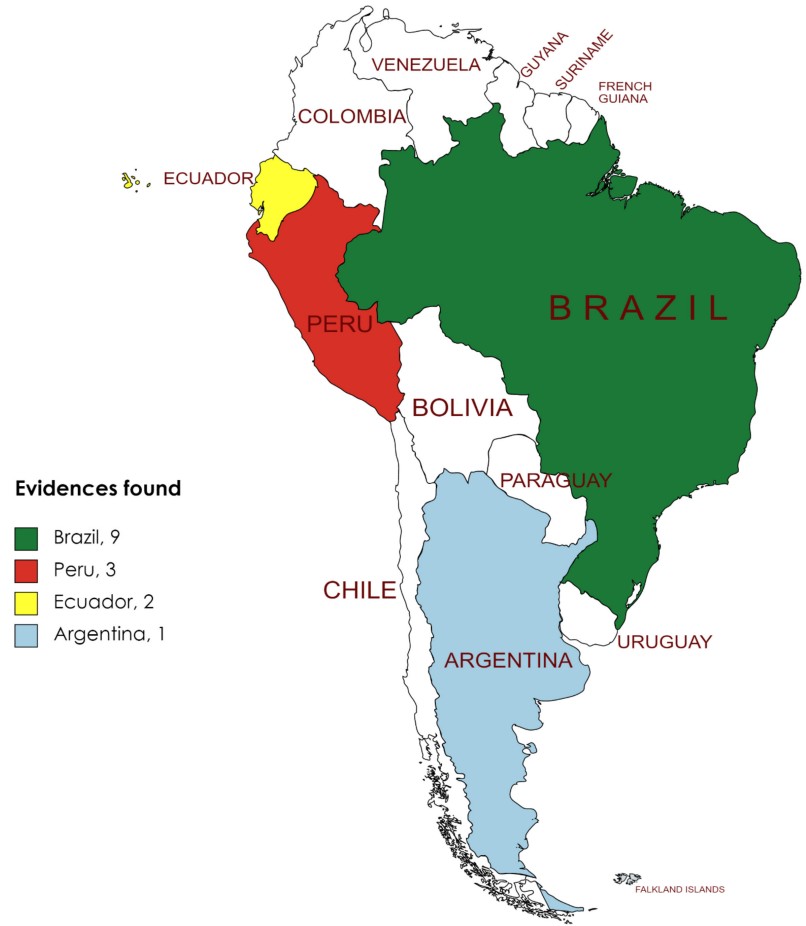

**Figure 2.** Country of origin for each record in South America.

Finally, Table 5 presents the conclusions of each article, accompanied by the lessons learned that can be drawn from such research.

**Table 5.** Conclusions and lessons for each article identified.

| Code | Conclusions | Lessons |
|------|-------------|---------|
| A1-1 | The community of Chulucanas, in Piura, Peru lives from tourism. Among the activities that generate the most work for the population is the sale of ceramic handicrafts. The art of working with ceramics is a family skill that is passed down from generation to generation, and that gives prosperity to the population by generating jobs. During the COVID-19 pandemic, the 12 family businesses that started the production of handmade ceramics since 1975 remained in business, despite little or no international or capital city tourist presence. The entrepreneurs who run these businesses are aware that this art must be preserved in order to perpetuate the renown of Chulucanas while continuing to provide jobs. | Sometimes, family businesses representative of a region tend to be more humane, reducing their profits in order to maintain the customs and traditions of the area. Nevertheless, they recognize that more tourism publicity is needed for the region, as well as more private investment to make it more attractive. They indicate that operating restaurants, hotels, recreation centers, a "Chulucanas" historic pottery museum, and pottery training workshops could help reactivate the economy and even create more jobs. |
| A1-2 | The women of the Association of Recyclable Materials practice social entrepreneurship, which prospects and implements innovative ideas and practical projects to achieve collective good. They received training on how to continue their work with optimal sanitary safety measures. The training was provided by a group of nurses committed to improving society. | Social entrepreneurship is a new archetype of development in networks and partnerships, with a focus on human, social and sustainable dimensions. In COVID-19 times they continued to work despite the health hazards because knew that about 30 families depended on their work. On the other hand, it is important to highlight the work of the nurses who, concerned about the health risks involved in working with recyclable waste, offered their help through a series of workshops. |
| A1-3 | High levels of resilience were identified in the 84 start-ups studied, which continued their activity in times of uncertainty and crisis. There were clear plans with effective task controls that increased the motivation of the members of the companies studied. | Work autonomy and organizational resilience are important drivers for promoting organizational resilience. It is advisable to reinforce these values in order to apply them in other contexts. |
| A1-4 | Entrepreneurs in the music sector have been hard hit by the collateral effects of the pandemic caused by COVID-19, such as the lockdown, the closure of entertainment venues, and the subsequent reduction of capacity at shows. The research reports, as support, that a previous study had a sample of 1910 observations (1293 solo musicians and 617 music groups). This study identified that 68% of the respondents admitted having reduced their income in times of pandemic. | Despite the cancellation of activities, the creative sector did not come to a standstill: both solo musicians (45.1%) and music groups (42%) developed new projects and products during the period of social withdrawal. 12% of individuals and 18.8% of organizations invested in the creation of revenue sources they had never used before, such as advance ticket sales and donation and/or crowdfunding campaigns. |
| A1-5 | Relational (name given by the authors for anonymity), started its business activities in 2014 as an educational blog. This entrepreneurship grew rapidly thanks to the right strategic decisions, but the year 2020 was crucial. As the global pandemic was declared, the company's partners came to a conclusion: it was essential to refocus their products. They moved away from creating large, process-intensive systems to offer smaller, more focused services to make it easier for customers to perceive value. The results obtained could not have been better, and plans for international expansion were drawn up. | In adverse scenarios, good planning and scenario management can help turn threats into growth opportunities. Today, Relational's case study serves as a guide for university students in courses related to decision making, marketing and the like. |
| A1-6 | The quantitative analysis of over 1000 entrepreneurs found that, when compared to adjacent nations, Ecuador has several lags that make it challenging to develop and maintain entrepreneurship over time, despite having substantial comparative advantages in terms of infrastructure and entrepreneurial ambition. The scenario indicated in this geographic area denotes regulatory constraints, restrictions for opening and shutting firms, online business, and innovation. For other types of self-employment, the economic uncertainty and lack of stability associated with precarious work prohibit people from viewing their job as a source of personal progress or well-being. However, the businesses analyzed continued to operate for a number of reasons. | From an economic point of view, entrepreneurs in South American countries such as Ecuador are accustomed to developing threat response plans. These same negative circumstances caused Manabi's entrepreneurs to reinvent themselves and manage to stay in the market and thus keep their workforce on the payroll. |

**Table 5.** *Cont.*

| Code | Conclusions | Lessons |
|---|---|---|
| A2-1 | The case study focuses on a supplier of organic products direct from the farm. While it is true that the procurement of plant products remained face-to-face throughout the pandemic, the offering to the final consumer did not. The measures proposed by the government to reduce the effects of the COVID-19 pandemic included the closure of commercial fairs (where they sold 100% of their merchandise), so they were forced to look for new forms of promotion and non-face-to-face sales. | The desire to continue offering quality products to the community is the main motivation for innovation. After the pandemic, the website was left used to make themselves known, Facebook and Instagram to promote, WhatsApp for orders and purchases (along with e-mail) and Excel for stock management. The business must go on, they say. |
| A2-2 | The authors of the study were able to interview 15 women entrepreneurs from different fields, obtaining as main results that women who venture into entrepreneurship usually share a role in the home while seeking to undertake entrepreneurship to improve their economy and face the crisis. It is concluded that women entrepreneurs have difficulties due to their limitations in terms of financing, their social role and gender stereotypes. In addition, women entrepreneurs show some concern about neglecting their home and family when starting a business; to this is added the lack of education and culture of medium and long-term planning, along with some deficiencies in the use of new technologies. It is essential to strengthen gender equality in entrepreneurship in order to overcome the problem of inequality in relation to men through training programs. | The entrepreneurs developed innovative solutions to continue their ventures, such as product innovation (decorative packaging, inclusion of cards with contact information, small gifts). There is also evidence of planning, such as having clear goals about what is expected from the business in the future. Similarly, the entrepreneurs developed negotiation and analytical thinking skills. |
| A2-3 | The bodegas in Peru are the main supplier of basic consumer goods to households within the traditional channel and have had a close relationship with their neighborhood neighbors since long ago. This trust translates into knowledge of the different activities in the neighborhood, granting weekly credit through the "fiado" mechanism, providing additional products through the "yapa", and being very cooperative and advisors to the neighbors. The COVID-19 pandemic created many challenges for the bodegas, which, with ingenuity and creativity with ingenuity and creativity, they were able to cope with the shortage of products, reduced seating capacity and limited opening hours. | The owners of the bodegas (bodegueros), as born entrepreneurs, have become digitalized through sales via social networks and mobile applications such as WhatsApp, and have delivered orders free of charge to their neighbors to avoid crowds in their stores. Thus, the "bodegueros" demonstrated their resilience in adapting to a sudden economic change, with unquantifiable consequences, such as the one caused by COVID-19. |
| A2-4 | Small and medium-sized enterprises were the most affected by the interruption of activities decreed by the Ecuadorian government to deal with the COVID-19 virus. Small businesses are vulnerable to situations such as the total paralysis of the productive apparatus, the normal mobilization of citizens which limits commercial exchange, as well as the high levels of contagion and mortality of their workers. | The results of the research show the adaptability and versatility that SMEs have, which led to the adoption of strategies such as the use of digital platforms or the redefinition of business models to ensure the survival of this important sector of the Ecuadorian economy; it is clear that the Ecuadorian State must develop policies to strengthen this sector of the economy by granting subsidies or special bank loans for this type of commercial organization. |
| A3-1 | The case study presents a fair held on a university campus, traditionally face-to-face. With the arrival of the pandemic and the regulations presented by the state and the university authorities, a climate of uncertainty initially arose, which was overcome through the use of technology, taking it to virtuality. | In the virtual environment there are greater advantages than in the face-to-face environment. Recorded presentations allow for rapid dissemination to the interested public, in real time or not. In addition, presenters can record their participation several times and then choose their best shot. Hybrid presentations are good alternatives in the context of the new normal. |

**Table 5.** *Cont.*

| Code | Conclusions | Lessons |
|------|-------------|---------|
| A3-2 | The COVID-19 pandemic motivated the institutions to make changes that can be explained by isomorphism, being focused on the business model, products and services, and customer relations. These changes have directly affected the banks' staff, who have become an important link for all these changes to take place. The information technology systems and the participation of these employees in the digital development processes presented evidence of how important digital entrepreneurship was for the institutions' objectives to be achieved. In addition, other benefits were perceived, such as operational efficiency to continue serving customers through other channels and customer confidence in these digital businesses. | Digital entrepreneurship is influenced in traditional banking institutions since the emergence of the COVID-19 pandemic. This is due to the fact that traditional banking institutions increasingly direct their products and services to the digital format, which allows them to contribute considerably to the direction and advancement of digital entrepreneurship. This movement allows one to understand the direct influence of the pandemic on this breakthrough, as it has taken traditional institutions out of their comfort zones. |
| A3-3 | The imposition of new ways of working has resulted in adaptation strategies that have brought gains for companies, which may become effective (for example, the home office). Other factors that reflect on the formulation and implementation of strategies can be mentioned, such as the change in the consumption pattern, and the insertion of new technologies. Pandemic has accelerated the use of digital media for sales, while allowing professional improvement. However, the evidence of this study does not allow generalizations, since heterogeneity in national economic structures and commercial networks may bring different consequences in each country. | Adaptation and resilience to the crisis occasioned by COVID-19, required the (re)design of strategies to overcome the impacts on business and individuals. The strategic and organizational processes for 'immediate response' denote the existence of dynamic capabilities. The moment of crisis can also broaden opportunities in relation to the insertion of new products and services and new customer niches. Resilience, strategic agility, and entrepreneurship will continue to play a key role in capturing value from opportunities and overcoming the crisis |
| A3-4 | As in the rest of the businesses related to the hotel, bar and restaurant industry, the case study presents the actions taken to stay afloat in times of uncertainty caused by COVID-19. In this case, permanent contact with regular customers was important. | Contact with the customer base should not be lost, although a site offering hospitality cannot be open to the public, it is possible to maintain revenue through pre-sales. In addition, new communication technologies allow managers to diversify and continue to bring revenue-generating options to their businesses. |
| A3-5 | The tourism sector was severely affected by the social isolation measures decreed by the government. The case study presents how a tourism blog was reinvented. Before the pandemic, the venture consisted of detailing travel experiences to later offer tourist packages to destinations. During the pandemic, they discovered that access to social networks globally increased by up to 40% in the world, including Brazil, which gave rise to a new market niche and a new occupation that they could explore, called "travel influencer". | It is possible that companies dedicated to the travel blog business diversify their activities and explore social networks. Applications such as TikTok, YouTube, among others, generate income for content creators who attract large audiences, thanks to original content such as travel experiences. |

## 4. Discussion

The COVID-19 pandemic has had a significant impact on the global economy, leading to bankruptcies of small businesses, job losses, and reduced working hours [88–93]. As a result, wages have also decreased due to the government's confinement policies [94–98]. However, some businesses have managed to thrive and even hire staff during these difficult times, thanks to their solid foundations and strategic planning [104].

The businesses that remained in operation have played a crucial role in sustaining the local economy and providing necessary goods and services to those in need [105,106]. This highlights the importance of entrepreneurship in the growth and development of society.

Entrepreneurship is a process that has had a positive impact on the establishment of enterprises over time, a phenomenon that leads to the creation of jobs, enhancing productivity efficiency, and achieving a degree of competitiveness in national and international markets [50,51].

Entrepreneurship is a methodological and systemic process that thrives even in hostile environments due to the attitudes of those who lead them, the actions they carry out, and their future aspirations, which allow them to realize their goals [25].

When talking about entrepreneurship, it is necessary to identify the person who carries out this activity [37,38]. While it is true that everyone has the potential to be an entrepreneur, not everyone can be one or successful in all scenarios [126,127]. In order to be an entrepreneur, it is important to identify opportunities, have knowledge of the activity to develop skills, not be afraid of failure, and always think with a positive attitude [37,38].

Thus, the globalized world requires people who develop the entrepreneurial process through their business ideas [128]. This requires the fulfillment of personal and professional requirements, as well as some essential factors that help the success of this process.

Technology has become a fundamental factor for new entrepreneurs who wish to enter the competitive world at a national and international level [57]. However, these entrepreneurs must be persistent and not be afraid of failure, as this usually accompanies every entrepreneur and the population in general. On the contrary, learn from experiences, look at the positive side, and keep in mind that this is part of the entrepreneurial process.

On the other hand, it is important to take into account entrepreneurship by necessity and by opportunity since these are influenced by the degree of development of the economic sectors of the countries.

## 5. Conclusions

South America has a high rate of entrepreneurship and self-employment. In this context, it was critical to find evidence and experiences that explain the reasons why entrepreneurs persisted during difficult times, such as the COVID-19 pandemic. This systematic review has yielded many relevant results that provide reflection for the academic community.

It is striking to note that a search for the word "entrepreneurship" (and its translations in Spanish and Portuguese) yielded only 783 initial results in the context of COVID-19. We believe that it is important to further explore this topic by publishing more scientific evidence in indexed journals.

Another important finding was that all the scientific evidence found under the searches for "*emprendimiento*" and "*empreendedorismo*" had previously been identified in the English language search. This only reflects that the requirement of high-impact South American journals to request both abstracts and *resumen/resumo* in articles pays off in terms of increased visibility.

In terms of the scientific evidence of entrepreneurship in South America during the COVID-19 pandemic, this systematic review revealed 15 records [111–119,121–125]. There were scientific articles that were written in one of three languages: Spanish (5 cases), Portuguese (3 cases), and English (4 cases); but there were also articles with two versions available: in Portuguese and English (3 cases). Although the authors of the correspondence are affiliated with a South American institution, the majority of their research was published in English, possibly to have a bigger impact and visibility. With the exception of 1, the journals in which the research was published were also South American. Similarly, the geographical settings in which the case studies were published in scientific journals were Brazil, Peru, Ecuador, and Argentina.

In terms of conclusions, it was discovered that all of the companies studied in the identified literature demonstrated resilience in the face of adversity. Despite the fact that the global COVID-19 pandemic prompted governments to enact measures such as the suspension of activities and, later, their limitation, these businesses have been continuing to operate in virtual mode, offering their services or developing plans for their return when it is possible to serve the public in person. In certain cases, the leaders of these businesses recognize that maintaining the economic stability of their employees and the community to which they belong is as important as maintaining the business itself.

The lessons learned focus more than anything on resilience, perseverance, and responsibility. Humanization on the part of organizational leaders and a desire to grow in an adverse environment could be found in each of the cases.

In the midst of the health crisis caused by the spread of COVID-19, where savings are scarce and revenues are lower than expected, only innovation and resilience can balance the gain and even put it in favor of the entrepreneur. The region's leaders are acutely aware of how important it is to stimulate the economy by establishing an enabling environment in which entrepreneurs can emerge, compete and innovate.

The difficulty in attaining accurate and thorough data, as well as inherent biases in self-reported data, was a research constraint in investigating entrepreneurship during the COVID-19 epidemic. Furthermore, due to the quickly changing nature of the pandemic and its influence on businesses, it was difficult to properly capture the repercussions in real time.

Another limitation of this systemic review was the difficulty of capturing relevant data. While it is true that Scopus contains the greatest number of high-quality scientific articles in social sciences, it has been discovered that many authors in the region continue to seek publication in South American scientific journals that are not necessarily indexed in Scopus. This is the reason why we also decided to include the Scielo database, which is one of the most relevant in the region.

As an academic implication, this study gives insights into how organizations adapt and respond to crisis situations, as well as how the pandemic has influenced entrepreneurial activity and success (with a focus on South America). This might help to shape future entrepreneurial research and teaching.

Socially, this study's findings might be utilized to support and enlighten government policy decisions on small company assistance during times of crisis. It might also give significant information to entrepreneurs, assisting them in navigating the pandemic's hurdles and guiding their decision-making. Furthermore, the results might be utilized to assist communities and give advice on how to effectively assist local businesses during times of crisis.

This study has practical implications as well. The findings of the review can help policymakers understand the challenges faced by entrepreneurs during the pandemic and design appropriate policies to support them. It can also provide insights into the strategies that have been successful for entrepreneurs during the pandemic, helping them to adapt to the changing business environment and survive in the long term.

Future studies on the theme should be conducted to increase the number of databases, including local databases (Redalyc and Latindex, for example) and relevant international databases such as the Web of Science. Additionally, studies could benefit from a more diverse sample, including both successful and unsuccessful entrepreneurs and those from other countries located in South America. Prior to that, it will be necessary to conduct a light search of these databases in order to assess the quality of the information.

**Author Contributions:** Conceptualization, V.H.F.-B. and J.d.J.S.G.-C.; methodology, V.H.F.-B.; data curation, M.E.M.-L.-R. and J.A.S.-V.; writing—original draft preparation, J.d.J.S.G.-C.; writing—review and editing, V.H.F.-B., M.E.M.-L.-R. and J.A.S.-V.; supervision, V.H.F.-B.; project administration, V.H.F.-B.; funding acquisition, V.H.F.-B. All authors have read and agreed to the published version of the manuscript.

**Funding:** This research was funded by Universidad César Vallejo, within the framework of the work plan outlined in RVI N° 052-2019-VI-UCV and The APC was funded by Universidad César Vallejo.

**Informed Consent Statement:** Not applicable.

**Data Availability Statement:** Data is contained within the article: Each article included in this systematic review is listed in the references section.

**Conflicts of Interest:** The authors declare no conflict of interest.

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
