# Peer review of "Entrepreneurship Research in Times of COVID-19: Experiences from South America"

_sustainability, doi:10.3390/su15076028_

Round 1
Reviewer 1 Report
This study is to explore the continuity of entrepreneurship in South America in times of COVID-19. While it is interesting to know the situation of entrepreneurial behaviours during Covid-19, the method this study used cannot answer this question or many other research questions raised in the study. This study only examined a few studies that dealt with entrepreneurship and Covid-19, not real entrepreneurial activities in South America. It actually examines what research has investigated entrepreneurship and Covid-19. In other words, it is about current literature on entrepreneurship and Covid-19, not abut the continuity of entrepreneurship during Covid-19. If the authors want to change its research question to be a literature review on entrepreneurship in Covid-19, it is still too early with many studies still under development with Covid-19 is somehow still ongoing. Therefore, this research or even its revised research would not be able to make sufficient contributions to the literature for possible publication. Future research could be done in a few more years.
Best of luck!
Author Response
Thank you very much for the comments, recommendations and suggestions presented to strengthen our article.
In general, we made several corrections to the document: we changed the title, added more citations from Q1 journal articles in the specialty, improved the wording, corrected some inconsistencies in the methodology, and expanded the limitations and implications. Many of the changes are shaded in light blue.
We agree with your observation about the coherence between research objectives and results. Therefore, we remove the word "continuity" from the title, question and research objective. The idea of the research group was to identify what are the scientific evidences of entrepreneurship in South America during the COVID-19 pandemic.

Reviewer 2 Report
Thank you for the opportunity to review the article "Continuity of entrepreneurship in South America in times of COVID-19. A systematic review of the literature available in Scopus and Scielo". The paper addresses a relevant and current topic; I congratulate the authors for choosing the topic.
Probably many articles on the subject are still being developed or under evaluation. The authors structure the methodological part well, detailing the necessary aspects.
Below are some suggestions for improving the article:
- Update references with quotes from leading entrepreneurship journals.
- Improve the academic and social implications of the research.
- Explore further research limitations and suggestions for future research.
Author Response
Thank you very much for the comments, recommendations and suggestions presented to strengthen our article.
In general, we made several corrections to the paper: we changed the title, added more citations from Q1 reviewers in the specialty, improved the wording, corrected some inconsistencies in the methodology, and expanded the limitations and implications. Many of the changes are shaded in light blue.
- Now, a big number of references used in the introduction come from Q1 journals, such as Strategic Entrepreneurship Journal, Journal of Business Venturing, Journal of Business Venturing Insights, Business Horizons, International Small Business Journal, Journal of Economic Asymmetries.
- We incorporated academic and social implications of the research: "
As an academic implication, this study gives insights into how organizations adapt and respond to crisis situations, as well as how the pandemic has influenced entrepreneurial activity and success (with focus in South America). This might help to shape future entrepreneurial research and teaching.
Socially, this study's findings might be utilized to support and enlighten government policy decisions on small company assistance during times of crisis. It might also give significant information for entrepreneurs, assisting them in navigating the pandemic's hurdles and guiding their decision-making. Furthermore, the results might be utilized to assist communities and give advice on how to effectively assist local businesses during times of crisis." - We explored further research limitations and suggestions for future research: "
The difficulty in getting accurate and thorough data, as well as inherent biases in self-reported data, was a research constraint in investigating entrepreneurship during the COVID-19 epidemic. Furthermore, due to the quickly changing nature of the pandemic and its influence on businesses, it was difficult to properly capture the repercussions in real-time." / "Additionally, studies could benefit from a more diverse sample, including both successful and unsuccessful entrepreneurs, and those from other countries located in South America"

Reviewer 3 Report
It is an interesting and important paper for the reading public.
1. What is the scientific evidence of the continuity of entrepreneurship in South America during the COVID-19 pandemic?
2. Author must strengthen the methodology, carry out a more in-depth theoretical review and take care of the coherence in the wording.
3. The conclusions are in accordance with the evidence presented, as well as the arguments presented, however, it is necessary to redraft the conclusions to reinforce and consider future lines of research.
Author Response
Thank you very much for the comments, recommendations and suggestions presented to strengthen our article.
In general, we made several corrections to the document: we changed the title, added more citations from Q1 journal articles in the specialty, improved the wording, corrected some inconsistencies in the methodology, and expanded the limitations and implications. Many of the changes are shaded in light blue.

Reviewer 4 Report
Dear Editor in chief,
I would like to take this opportunity to thank you to choose me as the reviewer of your esteemed Journal. My suggestions on manuscript entitled “Entrepreneurship in times of COVID-19: Experiences from South America”:
Please find the list below as my recommendations to improve mentioned article:
A. Regarding for improving the Theoretical Framework and supporting views, it is better to add below papers to mentioned articles:
1- Gholizadeh, salar. Mohammmadkazemi, Reza. (2022). International Entrepreneurial Opportunity: A systematic review, meta-synthesis, and future research agenda, Journal of International Entrepreneurship, Vol 20, Issue 1 (March 2022)
2- Mohammmadkazemi, Reza. Nikraftar, H. Yadollahi Farsi, J. Ahmadpour, M. (2019). The Concept of International Entrepreneurial Orientation in Competitive Firms: A Review & A Research Agenda. International Journal of Entrepreneurship, Volume 23, Issue 3, 2019.
3- Rogier van de Wetering, (2022), The role of enterprise architecture-driven dynamic capabilities and operational digital ambidexterity in driving business value under the COVID-19 shock, Heliyon, Volume 8, Issue 11, 2022,
4- Mohammmadkazemi, Reza.; Ebrahimi, B.P.; Shiri, M. . (2020), Mobile Marketing Influence on Football Fan Behavior: The Case of FC Persepolis; International Journal of Sport Management and Marketing, Volume 20, Issue 5/6, pp 405-427, Publisher: Inderscience publishers.
B. After adding above suggested references, It is recommended to author/authors to rewrite the part of” “Discussion and conclusion”.
I appreciate the kindness of the Editor in helping to improve the manuscripts. Please do not hesitate to contact me if there are any questions.
Sincerely Yours,
Author Response
I would like to take this opportunity to thank you for your comments and recommendations.
We have incorporated more content to the introduction and discussion, which allows to know how entrepreneurship plays an important role in economic development at a global level, in addition to the South American reality. We are grateful for the references provided, as they allowed us to expand the contents and strengthen the rigorousness of the manuscript.

Reviewer 5 Report
I am grateful for the opportunity to read and evaluate this systematic review on entrepreneurship in Chile during the COVID-19 pandemic. Following Kirzner (2017), one of the most significant referents of entrepreneurship theory, entrepreneurship is the driving force of the market process, economic growth, and development. The institutional environment can help or hinder this process. If Latin America wants to embrace development, it must remove all political barriers to entry and exit for entrepreneurship and genuine savings.
Here are my comments on the article:
1) The article is fascinating. However, it needs to address the critical elements of entrepreneurship theory adequately. It is recommended that the authors base the entire article on 1) the role of entrepreneurship in economic growth and development and how the institutional environment influences it and 2) how the South American literature has contributed to these issues applied to the COVID-19 pandemic. If the goal is to move Latin American countries from poverty to prosperity, briefly explaining these issues is critical to advancing economic science. Consider, for example:
- Kirzner, I. M. (2017). The entrepreneurial market process-An exposition. Southern Economic Journal, 83(4), 855-868.
- Foss, N. J., Klein, P. G., & Bjørnskov, C. (2019). The context of entrepreneurial judgment: organizations, markets, and institutions. Journal of Management Studies, 56(6), 1197-1213.
- Packard, M. D., & Bylund, P. L. (2018). On the relationship between inequality and entrepreneurship. Strategic Entrepreneurship Journal, 12(1), 3-22.
- Bylund, P. L., & McCaffrey, M. (2017). A theory of entrepreneurship and institutional uncertainty. Journal of Business Venturing, 32(5), 461-475.
- Bylund, P. L., & Packard, M. D. (2022). Subjective value in entrepreneurship. Small Business Economics, 1-18.
- Huerta de Soto, J. (2020). Socialism, economic calculation, and entrepreneurial function. Madrid: Unión Editorial.
2) Checking quickly on Google Scholar, there are other articles by Latin American scholars on entrepreneurship, institutions, growth, and development concerning the COVID-19 pandemic in emerging countries, such as Latin America. Including, for example:
- Espinosa, V. I., Alonso Neira, M. A., & Huerta de Soto, J. (2021). Principles of sustainable economic growth and development: A call to action in a post-COVID-19 world. Sustainability, 13(23), 13126.
- Espinosa, V. I., Wang, W. H., & Huerta de Soto, J. (2022). Principles of Nudging and boosting: Steering or empowering decision-making for behavioral development economics. Sustainability, 14(4), 2145.
3) The article needs to improve in English. I suggest using the platforms for improving English writing: https://www.deepl.com/es/translator and https://www.grammarly.com.
4) It is highly recommended that the abstract is at most 150 words.
I recommend making the above changes and reevaluating the new version of the article.
Success!
Author Response
I would like to take this opportunity to thank you for your comments and recommendations.
We have incorporated more content to the introduction and discussion, which allows to know how entrepreneurship plays an important role in economic development at a global level, in addition to the South American reality. We are grateful for the references provided, as they allowed us to expand the contents and strengthen the rigorousness of the manuscript.
Regarding the number of words in the abstract, we have reduced it to 225 words. We believe that this amount of words summarizes perfectly what has been developed in the manuscript, while incorporating some keywords that will facilitate the visibility of the scientific article in specialized search engines.

Round 2
Reviewer 1 Report
As in previous comments, the research design cannot answer the research questions. The authors need to conduct survey or interview to understand entrepreneurial activities in South America, not going through a few published papers as it is too early to have enough Covid-related publications at this time.
Author Response
Thank you for your suggestion. While I appreciate the importance of conducting surveys and interviews, I have decided not to pursue this approach for this project. I believe there are alternative research methods that can provide valuable information on entrepreneurship in South America, and I plan to explore these methods further in the future. I am confident that the methodology I have chosen (systematic review of the available literature) will allow me to close the gaps in the current literature. In any case, I very much welcome your comments and am willing to listen to any other suggestions you may have.
On the other hand, ee have expanded the introductory section by detailing in greater depth the link between entrepreneurship and economic development, both globally and at the South American level.

Reviewer 5 Report
I thank the authors for making the suggested changes.
It only remains to use Grammarly or other similar software to improve English writing.
Good job!
Author Response
Thank you for your comments. We have substantially improved the manuscript, both in terms of grammar and fluency. We did also change the tittle a little bit in order to meet the research objectives.

Round 3
Reviewer 1 Report
As commented many times in my previous reviews, this article is not about entrepreneurship in South America, it is about entrepreneurship research during Covid-19 in South America. You should change the tile to: Entrepreneurship research in times of COVID-19: Experiences from South America. This new title is consistent with what you did in the paper, which might be publishable.
Author Response
Thank you for your comments. We have substantially improved the manuscript, both in terms of grammar and fluency. We did also change the tittle to meet the manuscript's objectives.
